# First Report on *Colletotrichum fructicola* Causing Anthracnose in Chinese Sorghum and Its Management Using Phytochemicals

**DOI:** 10.3390/jof9020279

**Published:** 2023-02-20

**Authors:** Wei Zhao, Anlong Hu, Mingjian Ren, Guoyu Wei, Huayang Xu

**Affiliations:** College of Agriculture, Guizhou University, Guiyang 550025, China

**Keywords:** phytochemical, leaf spot, sorghum, *C. fructicola*, management

## Abstract

*Sorghum bicolor* is cultivated worldwide. Leaf spots on sorghum, which lead to leaf lesions and impaired growth, are prevalent and severe in Guizhou Province, Southwest China. In August 2021, new leaf spot symptoms were observed on sorghum plants growing in agricultural fields. We used conventional tissue isolation methods and pathogenicity determination tests. Inoculations of sorghum with isolate 022ZW resulted in brown lesions similar to those observed under field conditions. The original inoculated isolates were reisolated and fulfilled Koch’s postulates. Based on the morphological character and phylogenetic analyses of the combined sequences of the internal transcribed spacer (ITS) region and the β-tubulin (*TUB2*) and glyceraldehyde-3-phosphate dehydrogenase (*GAPDH*) genes, we identified the isolated fungus as *C. fructicola*. This paper is the first to report this fungus-causing disease in sorghum leaves. We studied the sensitivity of the pathogen to various phytochemicals. The sensitivity of *C. fructicola* to seven phytochemicals was measured using the mycelial growth rate method. Honokiol, magnolol, thymol, and carvacrol displayed good antifungal effects, with EC_50_ (concentration for 50% of the maximal effect) values of 21.70 ± 0.81, 24.19 ± 0.49, 31.97 ± 0.51, and 31.04 ± 0.891 µg/mL, respectively. We tested the control effect of the seven phytochemicals on the anthracnose caused by *C. fructicola*: honokiol and magnolol displayed good field efficacy. In this study, we expand the host range of *C. fructicola*, providing a basis for controlling sorghum leaf diseases caused by *C. fructicola*.

## 1. Introduction

*Sorghum bicolor* is an annual Poaceae with a high nutritional value. Its seeds contain 70% starch, 11% protein, vitamins (B_1_, B_2_, E), minerals, and micronutrients [1]. Sorghum has many applications, including brewing, food processing, and feed. Furthermore, this crop has strong resistance to stress, including drought, saline conditions, and high temperatures [2], and adapts to a wide range of soil types and pH values. Therefore, sorghum is grown worldwide and is cultivated in many regions of China. In 2018, sorghum planting accounted for 618,700 ha in China, according to the National Bureau of Statistics, with a production of approximately 290 million tons. However, the abusive use of pesticides, climatic variation, and other factors have resulted in the spread of several fungal diseases, some of which seriously impair the growth of sorghum plants, with leaf spot being one of the most damaging diseases that mainly harm leaves. The infected leaves develop an irregular chestnut spot, gradually expanding and causing severe necrosis. In China, the economic loss of sorghum due to leaf spot disease is approximately 12–55% [3]. Many pathogenic microorganisms are reported to cause sorghum disease, including *Pestalotiopsis trachycarpicola* [4], *Alternaria alternata* [5], *Curvularia clavata* Jain [6], *Drechslera australiensis* [7], *Fusarium thapsinum* [8], *Pantoea ananatis* [9], and yellow mosaic virus [10]. In recent years, sorghum disease has frequently occurred in sorghum-growing areas throughout China and tends to worsen each year, restricting the sustainable development of the sorghum industry.

*C. fructicola* has a wide geographic distribution and host range. It is one of the most difficult agricultural pathogens to control and causes disease in many crops, including hylocereus plants, culinary melon, tea, walnut, *Kadsura coccinea*, and *Bletilla striata* [11,12,13,14,15,16]. The fungus was first reported to cause diseases in sorghum leaves where sorghum plants with leaf spot disease were found in a sorghum plantation in Renhuai County, Zunyi City, Guizhou Province, China, with an incidence rate of approximately 21%. It severely limited photosynthesis, resulting in declines in sorghum yield and quality. In the early stage of infection, there are irregular brown spots that are white in the middle, round and oval leaves, and scattered small spots. These small spots link together to form larger spots, causing the leaves to wither. Identifying the cause of sorghum leaf spot disease and its control methods is critical to sorghum planting management. The use of phytochemicals to control plant diseases is one of the current research fields. For instance, carvacrol has been found to be effective in inhibiting the mycelial growth of three foliar pathogens (*Xanthomonas perforans*, *Alternaria tomatophila*, and *Podosphaeraxanthii*), and thymol has shown good antifungal activity against *Fusarium graminearum* due to the cell membrane damage originating from lipid peroxidation [17,18]. In this study, we aimed to determine the pathogenic factors of sorghum leaf spot disease and investigate potential phytochemical agents, considering the broader applications of these agents for agriculturally significant pathogens.

## 2. Materials and Methods

### 2.1. Diseased Leaf Collection, Fungus Isolation, and Phytochemicals

*Sorghum bicolor* (Hongyingzi) symptomatic leaves were collected from Renhuai County (27°6′ N, 106°4′ E) in May 2020. The sorghum leaves were first washed with sterile distilled water for 20 s to remove surface impurities. The leaves were disinfected with 75% (*v*:*v*) ethanol for 20 s and were washed three times with sterile distilled water. After surface disinfection, the leaves were cut into 0.5 cm square pieces and transferred to potato dextrose agar (PDA: potato infusion 200 g, glucose 20 g, agar 20 g, and distilled water 1 L) plates. After incubation at 28 °C for 2 days with 24 h of light, the colonies were transferred to new PDA plates and incubated at 28 °C for 5 days. Each isolate was inoculated with three plates. The pure colonies were soaked in 30% glycerol and stored at −80 °C for long-term storage. Shanghai Macklin Biochemical Co., Ltd. (Shanghai, China) provided phytochemicals (eugenol, magnolol, thymol, cinnamaldehyde, honokiol, carvacrol, geraniol) with purities of ≥98%, which were stored at 4 °C.

### 2.2. Pathogen Identification

We identified the isolates by morphology and DNA sequencing. We observed the morphology of the mycelia and fungal spores incubated at 28 °C for 10 days using an optical microscope (LEICA ICC50 W, Leica Microsystems Co., Ltd., Shanghai, China). The fungus morphology was identified as described in a previous study [19]. We extracted the genomic DNA of the pathogenic fungus using the Fungal Genomic DNA Extraction Kit (Tiangen Biotech, Beijing, China) as per the instructions. Universal primers targeting the ITS region, β-tubulin (*TUB2*), and glyceraldehyde-3-phosphate dehydrogenase (*GAPDH*) genes (Table 1) were used in the polymerase chain reaction (PCR) program to amplify the strain as per Watanabe’s method [20]. We performed amplification reactions with the primer pair for each gene in a Gcal cycler (T100TM Thermal Cycler, Bio-Rad, Hercules, CA, USA). Sangon Biotech Co., Ltd. (Shanghai, China) sequenced the amplified PCR products. Strain accession and GenBank accession were derived from NCBI’s GenBank nucleotide database (http://www.ncbi.nlm.nih.gov (accessed on 19 June 2022)), and a polygene phylogenetic tree was constructed using the maximum likelihood (ML) method in MEGA 7.0 software [21] with bootstrap values based on 1000 replications. *Monilochaetes infuscans* (CBS 896.96) was used as an outgroup.

### 2.3. Pathogenicity Assays

We isolated and used 14 isolates for fulfilling Koch’s postulates [24]. All the isolated strains were cultured in potato dextrose broth (PDB; 200.0 g of potatoes, 20.0 g of glucose, 1 L of water), shaken at 120 rpm, stored at 28 °C for 5 d, and filtered through gauze to collect the conidia. Then, 500 µL of 1 × 10^6^ conidia/mL was collected and sprayed on sterilized sorghum leaves with petioles. For a blank control, 500 µL of sterilized distilled water was sprayed. We placed each inoculated sorghum plant in a light incubator at 28 °C and 75% relative humidity with a 12/12 h light/dark photoperiod and regularly observed the disease progression of the leaves. We used three replicates for each isolate.

### 2.4. Antimicrobial Activity of Phytochemicals against Mycelial Growth

In order to identify phytochemicals with a good control effect on *C. fructicola*, we screened 11 phytochemicals. According to the mycelial growth rate method described by Xin [25], different phytochemicals were dissolved in appropriate organic solvents or water (geraniol and eugenol in ethanol; honokiol in dimethyl sulfoxide; cinnamaldehyde, carvacrol, and magnolol in acetone, and thymol in sterile water). All the phytochemicals were firstly dissolved in 20 µL of appropriate solvent, then diluted with water to prepare a series of concentration gradients. Then, 5 mL of solution was mixed evenly with 45 mL PDA medium. The *C. fructicola* colony (with a diameter of 6 mm) was placed in the center of the PDA medium containing phytochemicals and cultured at 25 °C and 75% relative humidity for 4 d under light conditions, and the colony diameter was then measured using a ruler. The EC_50_ (concentration for 50% of maximal effect) values of different phytochemicals were calculated using IBM SPSS analytics (SPSS Inc., Chicago, IL, USA) [26]. We performed all the experiments in triplicate.

### 2.5. Evaluations of Field Trials of Phytochemicals to Control the Disease Caused by C. fructicola

According to the survey, the disease began in early June. Field trials were conducted from April to August in 2021 and 2022 in Renhuai County (27°6′ N, 106°4′ E, elevation 880 m) in Guizhou Province, using a randomized complete block design with trial plots (16 m^2^) containing five treatments in three replicates [27]. The average values of nitrogen, phosphorus, potassium, organic matter, alkali-hydrolyzed nitrogen, available phosphorus, and available potassium in the soil were 1.74 g/kg, 0.75 g/kg, 19.90 g/kg, 30.90 g/kg, 100.28 mg/kg, 10.40 mg/kg, and 101.03 mg/kg, respectively. The disease in the experimental field was severe in the previous year. We soaked sorghum seeds in water for 24 h at 28 °C. Then, the seedlings were raised in greenhouses at 28 °C and 80% relative humidity with a 12/12 h light/dark photoperiod. After seven days, the seedlings were planted in a block with a size of 16 m^2^ with 20 × 20 cm spacing on 15 April 2021 and 15 April 2022, and sorghum seedlings were sown by hand at target depths of 3 cm and 5 cm. We observed that the sorghum leaves began to show disease spots in early June, and confirmed the symptoms caused by *C. fructicola* through the isolation and identification of the pathogen. Then, the phytochemicals were dissolved and configured to the appropriate concentration shown in below. Honokiol, magnolol, carvacrol, thymol, cinnamaldehyde, citronellol, geraniol, and eugenol were used in fields at concentrations of 21.7, 24.04, 31.04, 31.97, 41.17, 58.11, 59.96, and 89.47 µg/mL respectively. The prepared solution was evenly sprayed on the leaves using a CHNONLI compression sprayer (CHNONLI Co., Ltd., Shanghai, China) on 10 June 2021 and 15 June 2022. One month later, they were sprayed again according to the method above. We performed all the experiments in triplicate. The sorghum yield was measured. We assessed the disease severity (measured by the percentage of leaf area affected) as per Fehr et al. with minor modifications. [28]. One month after the second spray, we used a modified ‘X’ sampling pattern to select three random leaves per plot from the mid-canopy of the plants and three leaves from the top of the canopy to evaluate the disease severity caused by *C. fructicola*. These samples were photographed in the laboratory. We measured the spot area using ImageJ software (National Institute of Health, Bethesda, MD, USA) to determine the disease severity. The severity was calculated using the formula (A1 − A2)/A1 × 100, where A2 and A1 represent the scab and leaf areas [29], respectively. The control percentage was calculated using the formula (M1 − M2)/M1 × 100, where M1 and M2 represent the severity of disease in the control and treatment, respectively. We used a modified ‘X’ sampling pattern to select the plants and evaluated the disease incidence caused by *C. fructicola*. Disease incidence was calculated using the formula D1/D2 × 100, where D1 and D2 represent the number of diseased plants and the number of plants surveyed, respectively.

### 2.6. Statistical Analyses

All data analyses were performed using Excel 2010 and SPSS version 25 (SPSS Inc., Chicago, IL, USA). We performed one-way ANOVA as per Duncan’s multiple range test to determine the significance of differences (*p*-values < 0.05 were considered significant). We plotted charts with Origin 2021 and DPS.

## 3. Results

### 3.1. Isolation and Identification of Strain 022ZW from Sorghum Leaves

According to Koch’s postulates, 14 isolates were isolated and purified, and 1 × 10^6^ conidia/mL of the 14 isolates were inoculated into sorghum leaves. Only five isolates caused scabs, which were consistent with symptoms in the field, and the same fungi were isolated. The five strains were labeled as 019ZW (A), 020Z (B), 021ZW (C), 022ZW (D), and 023ZW (E). After incubation for five days, the sorghum leaves began showing symptoms. After 10 days, the lesion area was measured using ImageJ software (National Institute of Health, Bethesda, MD, USA). Isolate 022ZW from the PDA plate caused the largest size of spots. Based on the cultural and morphological characteristics of the colonies, we tentatively identified the pathogens as *Colletotrichum*. The characteristics of the colony and culture of the *Colletotrichum* species are shown in Figure 1. The shape and septum of the conidia are also described. The *Colletotrichum* colonies grew quickly (0.79 mm d^−1^ at 28 °C) and appeared white with developed aerial mycelia. When cultured for 14 days, the reverse side of the colony had brown particles, the reverse side had irregular blackspots, and the hyphae had compartments (Figure 2). The conidia were fusiform to slightly curved with rounded or elliptical ends, and the conidia were 14.8–23.2 × 2.4–3.8 µm in size (*n* = 50). The conidia were from the sporophore, with one or two septate (Figure 2). The morphological characteristics were consistent with published descriptions of *C. fructicola* [30].

For phylogenic analysis, we constructed a maximum parsimony tree using the combined sequences of ITS, *TUB2*, and *GAPDH* in MEGA 7.0 software with bootstrap values based on 1000 replications, as shown in Figure 3, including one species of *C. fructicola*, 18 other referred isolates of *Colletotrichum* species (Table 2), and an outgroup species (*Monilochaetes infuscans*). We aligned the sequences of representative isolates ITS, *TUB2*, and *GAPDH* to those of a different species of *Colletotrichum* obtained from NCBI’s GenBank nucleotide database. Phylogenetic analysis further confirmed that these strains (019ZW, 020ZW, 021ZW, 022ZW, 023ZW) and *C. fructicola* clustered together (Figure 3). Detailed morphological studies and sequence analyses confirmed that these strains belonged to *C. fructicola* (note: 019ZW, 020ZW, 021ZW, 022ZW, and 023ZW are all the pathogen *C. fructicola*). *Monilochaetes infuscans* (CBS 869.96) is the outgroup. We chose isolate 022ZW as the subject of our subsequent experiment because it caused the highest number of spots. The accession numbers of isolate 022ZW were ITS (OP523978), *GAPDH* (OP539309), and *TUB* (OP539308).

### 3.2. Phytochemical Sensitivity of Isolate 022ZW

Screening phytochemicals will be benefit to the development of green, low-toxicity fungicides that effectively control *C. fructicola.* The sensitivity testing results of the 11 phytochemicals against *C. fructicola* are shown in Table 3. Honokiol had a significant antibacterial activity with an EC_50_ value of 21.70 ± 0.81 µg/mL, followed by magnolol, thymol, carvacrol, citral, citronellol, geraniol, and eugenol, with EC_50_ values of 24.04 ± 0.49, 31.04 ± 0.89, 31.97 ± 0.51, 41.17± 0.69, 58.11 ± 0.28, 59.96 ± 1.21, and 89.47 ± 0.81 µg/mL, respectively. Other EC_50_ values of phytochemicals against *C. fructicola* are shown in Table 3.

### 3.3. Controlling Sorghum Leaf Spots Caused by C. fructicola Using Phytochemicals in the Field

Incidence of sorghum leaf spots were the same: 100% in different treatments. We identified leaf spots caused by *C. fructicola* through isolation and identification. Leaf spot disease can be significantly alleviated by the spraying of phytochemicals. Honokiol and magnolol showed good treatment effects, whereas those of carvacrol, thymol, citral, citronellol, and geraniol were less effective. The control effects of eugenol were poor. All the phytochemicals had control effects, and all the treatments boosted production compared with the control. The spraying of honokiol and magnolol effectively increased production (Table 4).

## 4. Discussion

Sorghum (*Sorghum bicolor*) is native to Africa and China [31], which have a long history of cultivating the fifth most widely consumed cereal in the world [32]. Sorghum has a wide range of uses [33]. In addition to food use, it is used for brewing, sugar, pharmaceuticals, bioenergy purposes, etc. Sorghum can tolerate drought, saline conditions, and high temperatures [2]. Sorghum is of great economic importance to China, where production reached approximately 290 million tons in 2021/22. With the gradual increase in sorghum-planting areas worldwide, pathogenic fungi causing sorghum leaf disease have been frequently reported, including *Ramulispora sorghi*, *Pestalotiopsis trachycarpicola*, *Gloeocercospora sorghi, Colletotrichum graminicola, Exserohilum turcicum, Alternaria alternata*, *Cercospora fusimaculans*, and *Cercospora sorghi*. We conducted conventional methods of tissue isolation and pathogenicity determination in this study. Fourteen strains were isolated, and five of these isolates caused leaf necrosis and irregular lesions. The five isolates were identified as *C. fructicola*. *C. fructicola* is also an important pathogenic factor of anthracnose disease in mango [34], watermelon anthracnose [35], apple bitter rot [36], pear bitter rot [37], chili anthracnose [38], and strawberry crown rot disease [39]. However, climatic variation, human activity, and other factors may cause fungi to experience host jumping in plants [40]. Our study is the first to report on *C. fructicola* causing anthracnose in sorghum. Other researchers should consider its impact and measures for forecasting it, and management practices should be implemented.

At present, chemical management is the most important means of protecting crops and treating diseases caused by fungus. For example, carbendazim and propiconazole could prevent anthracnose, grey leaf spot and zonate leaf spot on sorghum; carbendazim could also control wet root rot on sorghum [41]; carbendazim effectively inhibited the hypha of *C. fructicola* [30]; and *C. fructicola* were highly sensitive to pyraclostrobin, difenoconazole, fludioxonil, tebuconazole, pyrisoxazole, and tetramycin in terms of mycelial growth inhibition [42]. However, the unscientific and irrational use of chemical pesticides results in residues, resistance, environmental pollution, and other problems [43]. For instance, benzimidazole fungicides reside in soil; benomyl metabolite carbendazim has reproductive toxicity; and Usman et al. found that the long-term use of procymidone and fludioxonil could lead to increased resistance [44]. Chechi et al. found that if these fungicides were not applied scientifically, they could induce resistance in *C. fructicola* [45,46,47]. Natural agents have attracted attention because phytochemicals have advantages that can potentially control plant diseases. They are low in toxins, do not easily lead to pesticide resistance, and meet the criteria for IPM and organic farming [48]. Carvacrol can control vegetable diseases [49], and 2-allylphenol (2-AP) is an excellent fungicide against many plant pathogens [50]. It has been reported that cinnamaldehyde inhibits *F. sambucinum* ergosterol biosynthesis [51], and that magnolol significantly damages the plasma membrane of *Rhizoctonia solani* [26].

We screened 11 phytochemical agents for antifungal activity and determined that honokiol, magnolol, carvacrol, thymol, citral, citronellol, geraniol, and eugenol inhibited the growth of *C. fructicola*. We applied eight inhibitory phytochemicals with better inhibition effects to the field and found that honokiol and magnolol effectively controlled the disease and increased yield, whereas eugenol had the worst control effect, and the other phytochemicals had an average effect. The mechanisms of these phytochemical agents’ antifungal effects are still unclear and require further study.

## 5. Conclusions

In this study, we determine that the strains 019ZW, 020ZW, 021ZW, 022ZW, and 023ZW–which we identified as *C. fructicola* by morphological characteristics, molecular biology, and pathogenicity verification–cause leaf spot disease in sorghum. We screened 11 phytochemical agents, and found that honokiol, magnolol, carvacrol, and thymol had the potential to inhibit mycelium growth. Honokiol was the most effective against *C. fructicola* in vitro. We applied eight phytochemicals with the better inhibition effects to the field, and found that honokiol and magnolol could effectively control the disease and increase yield.

## Figures and Tables

**Figure 1 jof-09-00279-f001:**
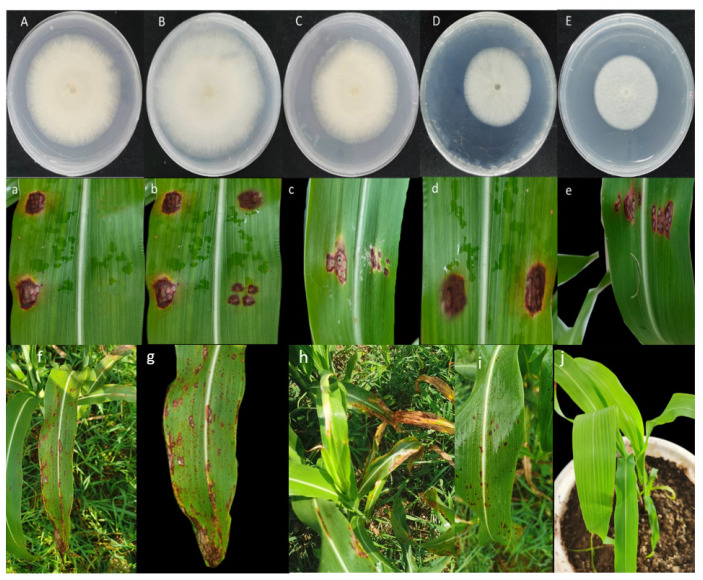
Pathogenicity assays. Five isolates caused spots consistent with symptoms in the field. (**A**–**E**) correspond to (**a**–**e**); (**f**–**i**) symptoms in the field; (**j**) control.

**Figure 2 jof-09-00279-f002:**
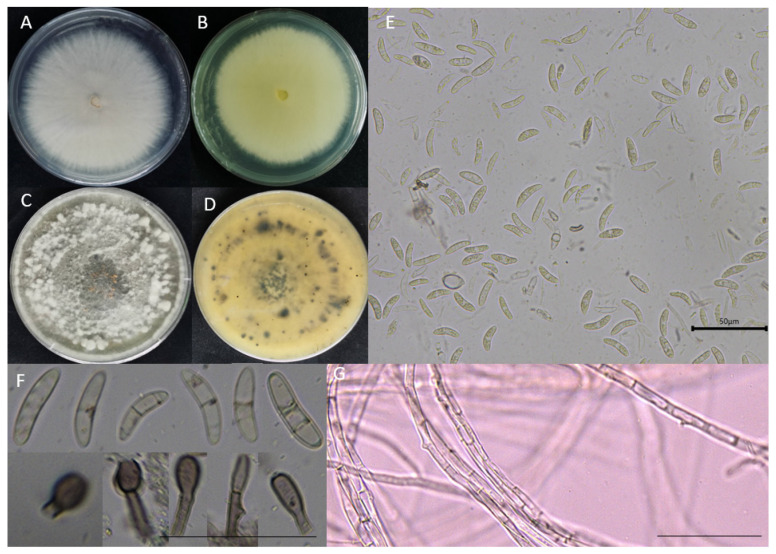
The morphology of mycelia and fungal spores grown on PDA medium for 14 days. (**A**,**C**) Front of the colony; (**B**,**D**) back of colony; (**E**,**F**) conidia and sporophore; (**G**) hyphae. The scale bars in Figure 2E–G are all 50 µm.

**Figure 3 jof-09-00279-f003:**
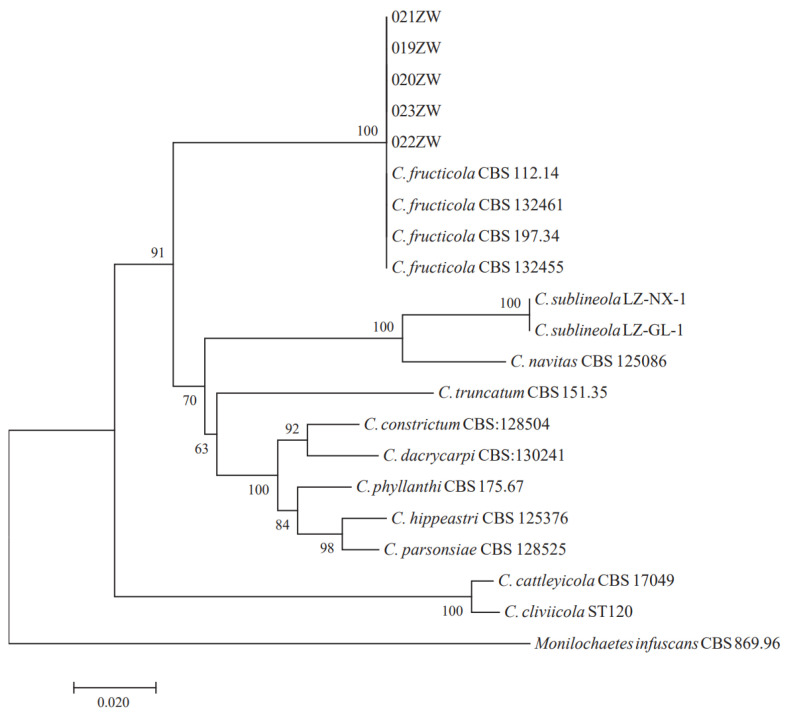
The ITS–*GAPDH–TUB2* phylogenetic tree. 019ZW, 020ZW, 021ZW, 022ZW, 023ZW, and *C. fructicola* clustered together.

**Table 1 jof-09-00279-t001:** PCR primers for ITS, *GAPDH,* and *β-TUB* amplification.

Target Sequence	Primer	Primer Sequence (5′→3′)
ITS	ITS1	TCCGTAGGTGAACCTGCGG
	ITS4	TCCTCCGCTTATTGATATGC
*GAPDH* [22]	*GDR*	GGGTGGAGTCGTACTTGAGCATGT
	*GDF*	GCCGTCAACGACCCCTTCATTGA
*TUB2* [23]	*Bt2a*	GGTAACCAAATCGGTGCTGCTTTC
	*Bt2b*	ACCCTCAGTGTAGTGACCCTTGGC

**Table 2 jof-09-00279-t002:** Reference isolates used in the present study and their GenBank accession numbers.

Species	Strain Accession	GenBank Accession		
		ITS	GAPDH	TUB2
*C. fructicola* *C. cattleyicola*	022ZW CBS 17049	OP523978 MG600758	OP539309 MG600819	OP539308 MG601025
*C. cliviicola*	ST120	MH291214	MH291258	MH458027
*C. navitas*	CBS 125086	JQ005769	–	JQ005853
*C. sublineola*	LZ-NX-1	MK881657	MK881674	MK881725
*C. cliviicola*	LZ-JY-1	MK881658	MK881675	MK881726
*C. sublineola*	LZ-GL-1	MK881659	MK881676	MK881727
*C. fructicola*	CBS 112.14	KC566786	KC566640	KC566208
*C. fructicola*	CBS 132461	KC566784	KC566638	KC566206
*C. fructicola*	CBS 197.34	KC566789	KC566643	KC566211
*C. fructicola*	CBS 132455	KC566788	KC566642	KC566210
*C. cymbidiicola*	CBS 128504	JQ005238	JQ005325	JQ005672
*C. cymbidiicola*	CBS 130241	JQ005236	JQ005323	JQ005670
*C. hippeastri*	CBS 125376	JQ005231	JQ005318	JQ005665
*C. phyllanthi*	CBS 175.67	JQ005221	JQ005308	JQ005655
*C. torulosum*	CBS 151.35	GU227862	GU228254	GU228156
*C. parsonsiae*	CBS 128525	JQ005233	JQ005320	JQ005667
*Monilochaetes infuscans*	CBS 869.96	JQ005780	JX546612	JQ005864

Note: ‘–’ indicates that there are no GAPDH genes in the GenBank accession.

**Table 3 jof-09-00279-t003:** Antimicrobial activity of phytochemicals against strain 022ZW.

Phytochemicals	Concentration (µg/mL)	Regression Equation	EC_50_ (µg/mL)	Coefficient of Determination (R^2^)	95% Confidence Interval
Honokiol	400, 200, 100, 50, 25	y = 2.9817 + 1.5100x	21.70 ± 0.81	0.9904	13.68–34.44
Magnolol	300, 200, 100, 50, 25	y = 2.5091 + 1.8038x	24.04 ± 0.49	0.9980	15.41–37.48
Carvacrol	150, 100, 50, 20, 10	y = 2.1503 + 1.9100x	31.04 ± 0.89	0.9256	18.41–52.34
Thymol	500, 200, 100, 50, 25	y = 2.6682 + 1.5495x	31.97 ± 0.51	0.9744	17.64–57.96
Citral	500, 200, 100, 50, 25	y = 1.0774 + 1.7119x	41.17 ± 0.69	0.9129	28.53–89.76
Citronellol	60, 30, 20, 10, 5	y = 1.9081 + 1.7525x	58.11 ± 0.28	0.9835	36.09–93.56
Geraniol	150, 100, 50, 20, 10	y = 3.4124 + 0.8930x	59.96 ± 1.21	0.9887	25.31–142.01
Eugenol	400, 200, 100, 50, 25	y = 3.1670 + 0.9392x	89.47 ± 0.81	0.9835	18.77–426.31
Citronellal	80, 40, 20, 10, 5	y = 1.3826 + 1.4634x	296.39 ± 0.11	0.9349	133.49–658.07
Cinnamaldehyde	500, 200, 100, 50, 25	y = 1.8266 + 1.1433x	596.69 ±1.10	0.9639	214.57–1659.30
Resveratrol	500, 200, 100, 50, 25	y = 2.9644 + 0.6950x	849.37 ± 0.30	0.9930	637.04–1264.24

**Table 4 jof-09-00279-t004:** Effects of phytochemical treatments on disease control and yield.

Phytochemical Treatment	Year					
	2021	2022	2021	2022	2021	2022
	Sorghum Yield (kg/16 m^2^)		Severity (%)	Severity (%)	Control Percentage %	Control Percentage %
Honokiol	13.33 ± 0.16 A	14.05 ± 0.41 A	6.61 F	5.66 E	70.45%	74.54%
Magnolol	13.41 ± 0.21 A	14.19 ± 0.11 AB	6.04 EF	6.01 DE	72.95%	72.96%
Carvacrol	12.44 ± 0.35 B	13.03 ± 0.12 BC	8.15 DE	6.97 CED	63.52%	68.65%
Thymol	12.23 ± 0.13 B	12.83 ± 0.72 CD	9.23 CD	7.38 CD	58.71%	66.80%
Citral	10.95 ± 0.17 C	11.71 ± 0.22 DE	9.68 CD	7.99 C	56.69%	64.06%
Citronellol	11.41 ± 0.37 C	11.93 ± 0.35 CDE	10.52 C	7.51 CD	52.91%	55.83%
Geraniol	11.02 ± 0.29 C	11.54 ± 0.43 EF	10.95 C	7.56 CD	51.02%	53.71%
Eugenol	10.05 ± 0.46 D	10.51 ± 0.22 F	16.13 B	11.37 B	27.82%	36.93%
Control	8.23 ± 0.29 E	8.55 ± 0.21 G	22.35 A	22.23 A	-	-

Numerical values were expressed as mean ± standard error (SE) of triplicates. Different uppercase letters represented a significant difference (*p* < 0.05).

## Data Availability

The datasets generated and/or analyzed in the study are available from the corresponding author upon reasonable request.

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
