# Peer review of "First Report on Colletotrichum fructicola Causing Anthracnose in Chinese Sorghum and Its Management Using Phytochemicals"

_jof, 2023, doi:10.3390/jof9020279_

Round 1

Reviewer 1 Report

In this work, the authors are identifying for the first time Colletotrichum fructicola  as the causal agent of anthracnose on Chinese sorghum.

The fungal isolates obtained were studied by means of morphological, cultural and molecular characterization. Pathogenicity  tests have allowed fulfilling the Koch’s Postulates for this disease.

Below you will find my changes and suggestions for the authors:

Line 70: Please explain what “full exposure” means.

Line 71: Please explain why you repeated this step three times. Are the  cultures monosporic?

Line 130: If the field trials were conducted between April to August in 2021 and 2022, why were the phytochemical solutions sprayed in 2020 and 2021?

 Line 146: Delete “A phylogenetic tree was constructed using MEGA 7.0”. It is not a statistical analysis.

Line 256: What does it mean that chemical fungicides are not scientific? Please explain.

In the discussion part on fungicides, the part on the current use of fungicides for Colletotrichum anthracnose should be expanded.

Author Response

Dear reviwer

Thank you for your valuable comments on my article. The Manuscript ID is jof-2181680. The title of this manuscript is First Report on Colletotrichum fructicola Causing Anthracnose in Chinese Sorghum and its management using phytochemicals.” I have answered all your comments.

Line 70

Point 1:Please explain what “full exposure” means.

Response 1: This is my writing error, the “full exposure” has been changed to “24 h of light”. (Line 72)

Line 71

Point 2:Please explain why you repeated this step three times. Are the  cultures monosporic?

Response 2: It is not cultures monosporic. The purpose of repeating it three times is to prevent other fungal and bacterial contamination. The sentence has changed to “Each isolate was inoculated with three plates” (Line 74)

Line 130

Point 3:If the field trials were conducted between April to August in 2021 and 2022, why were the phytochemical solutions sprayed in 2020 and 2021?

Response 3: This is my writing error. 2020 and 2021 has been changed to 2021 and 2022

(Line 128)

Line 146

Point 4: Delete “A phylogenetic tree was constructed using MEGA 7.0”. It is not a statistical analysis.

Response 4: I have deleted this sentence. (Line 167)

Line 256

Point 5: What does it mean that chemical fungicides are not scientific? Please explain.

In the discussion part on fungicides, the part on the current use of fungicides for Colletotrichum anthracnose should be expanded.

Response 5: This is a misrepresentation, “chemical fungicides are not scientific” has been changed to“unscientific and irrational use of chemical fungicides”.

    There are expansion about current use of fungicides for Colletotrichum anthracnose

“arbendazim was effectively inhibited the hypha of Colletotrichum fructicola, and Colletotrichum fructicola were highly sensitive to pyraclostrobin, difenoconazole, fludioxonil, tebuconazole, pyrisoxazole, and tetramycin in terms of mycelial growth inhibition. Usman et al. study and find that long-term use of procymidone and fludioxonil can lead to increased resistance”. (Line 274)

Reviewer 2 Report

Here are some suggestions that could improve the text.

 Lines 22-23: “We tested the control effect of 22 the seven phytochemicals on anthracnose caused by Colletotrichum fructicola.” > it was already said three lines ago.

Line 35: “ hm2”    > hectares   ? > ha

Line 43: “ Jain” > this is the name of whoever described the species. Either it is used for all species (the first time they are mentioned in the text) or it is never used. It should also not be written in italics

Line 63 and 73: “Phytochemical Agents” > “phitochemigals” “ fungicides” “agrochemicals”  better than “Phytochemical Agents”

Line 64”: “pathogenic …leaves”    > “symptomatic leaves”

Line 68: “0.5 mm square”  >  how did the authors cut square pieces of only 0.5 millimeters (sideways, I assume). If they were 0.5 mm square, it would be even more difficult to cut a square with a side of 0.7 mm ....... verify the data.

Line 69: “200.0 g of potato” > 200 would be the grams of potato used to produce the infusion needed for a liter of medium. The sentence as written is misleading.

Line 70: “full exposure” > ?

Line 88: “We constructed a polygene phylogenetic tree” > it would be useful to describe how the data were assembled and not to mention only the software used.

Line 98: “Koch’s hypothesis” > “fulfilling Koch’s postulates”

Line 101: “6”   > should be written as a superscript

Line 110: “thymol in sterile water”   >water it is not an organic solvent

Lines 110-111: “Then, water with the corresponding concentration of sol-110 vents was mixed evenly with the PDA medium at different concentrations.” > it is not clear at what concentrations the solvents were used nor has it been stated what this part of the experiment is for.

Lines 106-114: I understand the importance of reducing sentences and shortening texts... but the description of this test should be implemented.

Line 115: I would completely change the title of the paragraph. I would completely change the title of the paragraph by inserting the concept that we are dealing with evaluations of field trials of fungicides

Line 119: “We soaked” > into what (I guess water)? I assume that later the seeds were sown: where, how.

Line 123: “the disease began in early June” > at this point it is assumed that it is a natural infection. Perhaps saying it earlier, when the field was mentioned, would have been useful.

Lines 123-124: “We observed that the sorghum leaves began to show disease spots in early June through isolation and identification” > Through isolation and identification (of what?) you don't start to see the disease. At the very least you confirm the pathogenic nature of a strain. The sentence should be rephrased by separating when symptoms were seen from pathogen isolation.

Line 128: “in 21.7, 24.04, 31.04, 31.97, 41.17, 58.11, 59.96, 89.47 respectively” >  the unit of measurement relating to the numbers just listed is completely missing. How can an experiment described in this incomplete way be replicated?

Lines 132-133: “We assessed disease severity (measured by the percentage of leaf 132 area affected) as per Fehr et al. [24].” >  it is fine to insert bibliographic references of the methods but it is not pleasant not to have even a hint of the method itself.  It would also be appropriate to describe the assessment of both disease severity and disease incidence. different parameters but equally useful to describe the effect of a fungicide.

Line 150: “106” > pay attention how the number is written.

Line 154: “disease spots” > it is better to describe the symptom

Lines 155-156: “(National Institute of Health, Bethesda, MD, USA).”  > this should be described in the methods not in te result’s chapter.

Line 157: “amount” > number, size or both?

Line 158: “app.” > ?

Line 162: “obverse” > “reverse”

Lines 181-183: “We amplified the ITS, GAPDH, and TUB2 sequences using PCR ……. software.” > metods not results.

Line 189:” The accession numbers of isolate…” > The accession numbers are retated to sequences of isolates…

Line 206: “C. fructicola” > when the text is in italics, latin name should be written normally.

Lines 207-208: why repeat these sentences?

Lines 209-211: this sentences lack of something….. these are the concentration that showed a protection or the used? It should be specify.

Line 241: “Weconducted” > “We conducted”

Lines 249-250: “There have been no reports on diseases caused by Colletotrichum fructicola in sorghum.” > It is useless: the same concept was illustraded the phrase before.

Lines 253-261: “For example……” > it is better to cite the use of fungicides on sorghum.

Table 4: Control percentage(%)> it is obvious that a control percentage was expressed in percentage > remove parenthesis

References: the titles of cited articles should be writte according the same rules….. uppercase/lowercase for the initials letters….

Be careful to the many latin names e.g. line 303 “pestalotiopsis trachycarpicola  >  Pestalotiopsis trachycarpicola  (also in italics),

also the journals…. Some in italics some not…. Please follow the instruction for the Authors

 Also, in some cases, the English text should be revised.

Author Response

Dear reviewer

Thank you for your valuable comments on my article. The Manuscript ID is jof-2181680. The title of this manuscript is First Report on Colletotrichum fructicola Causing Anthracnose in Chinese Sorghum and its management using phytochemicals.” I have answered all of your comments.

Line 22-23

Point 1:  “We tested the control effect of the seven phytochemicals on anthracnose caused by Colletotrichum fructicola.” > it was already said three lines ago.

Response 1: This is an experiment in the field. I have changed the sentence to “We tested the control effect of the seven phytochemicals on anthracnose caused by Colletotrichum fructicola, honokiol and magnolol displayed good field efficacy”. (Line 23 )

Line 35

Point 2: “ hm2”    > hectares   ? > ha

Response 2:  hm= hectares = ha. hm2 has been changed to hectares.  (Line 36 )

Line 43

Point 3: “ Jain” > this is the name of whoever described the species. Either it is used for all species (the first time they are mentioned in the text) or it is never used. It should also not be written in italics

Response 3: The“ Jain” has been changed to Jain.  (Line 45 )

Line 63 and 73

Point 4: “Phytochemical Agents” > “phitochemigals” “ fungicides” “agrochemicals”  better than “Phytochemical Agents”

Response 4: The “Phytochemical Agents” has been changed to phitochemigals  (Line 67 and 76 )

Line 64

Point 5: “pathogenic …leaves” > “symptomatic leaves”

Response 5: The “pathogenic …leaves” has been changed to “symptomatic leaves” (Line 66 )

Line 68

Point 6: “0.5 mm square”  >  how did the authors cut square pieces of only 0.5 millimeters (sideways, I assume). If they were 0.5 mm square, it would be even more difficult to cut a square with a side of 0.7 mm ....... verify the data.

Response 6: This is a writing error, “0.5 mm square” has been changed to“0.5 cm square” (Line 70)

Line 69

Point 7: “200.0 g of potato” > 200 would be the grams of potato used to produce the infusion needed for a liter of medium. The sentence as written is misleading.

Response 7: “200.0 g potato” has been changed to 200.0 g potato/L. PDA: 200 g potatoes/L, 20 g glucose/L, 20 g agar/L, and 1 L distilled water (Line 71 )

Line 70

Point 8: “full exposure” > ?

Response 8: This is my writing error, the “full exposure” has been changed to “24 h of light”.

(Line 72 )

Line 88

Point 9: “We constructed a polygene phylogenetic tree” > it would be useful to describe how the data were assembled and not to mention only the software used.

Response 9: The “strain accession and GenBank accession were derived from NCBI’s GenBank nucleotide database(http://www.ncbi.nlm.nih.gov)” was added, and the “in MEGA 7.0 software” was removed.  (Line 91 )

Line 98

Point 10: “Koch’s hypothesis” > “fulfilling Koch’s postulates”

Response 10: “Koch’s hypothesis” has changed to “fulfilling Koch’s postulates”  (Line 103 )

Line 101

Point 11: “6”   > should be written as a superscript

Response 11: “6”  has been written as a superscript (Line 106 )

Line 110

Point 12: “thymol in sterile water” >water it is not an organic solvent

Response 12: The “or water” was added. different phytochemicals were dissolved in appropriate organic solvents or water (geraniol and eugenol in ethanol; honokiol in dimethyl sulfoxide; cinnamaldehyde, carvacrol, and magnolol in acetone; thymol in sterile water ).  (Line 115 )

Lines 110-111

Point 13: “Then, water with the corresponding concentration of solvents was mixed evenly with the PDA medium at different concentrations.” > it is not clear at what concentrations the solvents were used nor has it been stated what this part of the experiment is for.

Response 13: All phytochemicals were firstly dissolved in 20 µL of appropriate solvent, then diluted with water to prepare a series of concentration gradients. Then, 5mL of solution were mixed evenly with 45 mL PDA medium. The Colletotrichum fructicola colony (with a diameter of 6 mm) was placed in the centre of the PDA medium containing phytochemicals and cultured at 25 â—¦C and 75% relative humidity for 4 d under light conditions, then the colony diameter (mm) was measured using a ruler. This purpose of this part experiment was increase. “In order to screen out phytochemicals with good control effect on Colletotrichum fructicola”, (Line 117- 123 )

Lines 106-114

Point 14: I understand the importance of reducing sentences and shortening texts... but the description of this test should be implemented.

Response 14: In order to screen out phytochemicals with good control effect on Colletotrichum fructicola, according to the mycelial growth rate method described by Xin [23], different phytochemicals were dissolved in appropriate organic solvents or water (geraniol and eugenol in ethanol; honokiol in dimethyl sulfoxide; cinnamaldehyde, carvacrol, and magnolol in acetone, and thymol in sterile water ). All phytochemicals were firstly dissolved in 20 µL of appropriate solvent, then diluted with water to prepare a series of concentration gradients. Then, 5mL of solution were mixed evenly with 45 mL PDA medium. The Colletotrichum fructicola colony (with a diameter of 6 mm) was placed in the centre of the PDA medium containing phytochemicals and cultured at 25 ℃ and 75% relative humidity for 4 d under light conditions, then the colony diameter (mm) was measured using a ruler. The EC50 (concentration for 50% of maximal effect) values of different phytochemicals were calculated using IBM SPSS analytics (SPSS Inc., Chicago, IL, USA) [24]. We performed all experiments in triplicate. (Line 113,117- 123 )

Lines 115

Point 15:  I would completely change the title of the paragraph. I would completely change the title of the paragraph by inserting the concept that we are dealing with evaluations of field trials of fungicides

Response 15: the title has been changed to “Evaluations of Field Trials of phytochemicals to Colletotrichum fructicola” (Line 126 )

Line 119:

Point 16: “We soaked” > into what (I guess water)? I assume that later the seeds were sown: where, how.

Response 16: where: We soaked sorghum seeds into water. Field trials were conducted from April to August in 2021 and 2022 in Renhuai County, (27°6′ N, 106°4′ E, elevation 880 m) Guizhou Province. Soil fertility is added > “The average values of nitrogen, phosphorus, potassium, organic matter, alkali-hydrolyzed nitrogen, available phosphorus and available potassium in soil were 1.74 g/kg, 0.75 g/kg, 19.90 g/kg, 30.90 g/kg, 100.28 mg/kg, 10.40 mg/kg and 101.03 mg/kg , respectively”. How: sorghum seedings were sown by hand at target depths of 3 cm and 5 cm. (Line  127, 131-133 )

Line 123

Point 17: “the disease began in early June” > at this point it is assumed that it is a natural infection. Perhaps saying it earlier, when the field was mentioned, would have been useful.

Response 17: The sentence “According to the survey, the disease began in early June. was put at the beginning of the paragraph. (Line 127  )

Lines 123-124

Point 18: “We observed that the sorghum leaves began to show disease spots in early June through isolation and identification” > Through isolation and identification (of what?) you don't start to see the disease. At the very least you confirm the pathogenic nature of a strain. The sentence should be rephrased by separating when symptoms were seen from pathogen isolation.

Point 18: We observed that the sorghum leaves began to show disease spots in early June , and confirmed that the sympotm were cuaseed by Colletotrichum fructicola through isolation and identification of pathogen. (Line 139 )

Line 128

Point 19: “in 21.7, 24.04, 31.04, 31.97, 41.17, 58.11, 59.96, 89.47 respectively” >  the unit of measurement relating to the numbers just listed is completely missing. How can an experiment described in this incomplete way be replicated?

Response 19: I've already added the units (µg/mL). 21.7, 24.04, 31.04, 31.97, 41.17, 58.11, 59.96, 89.47 µg/mL.  (Line 145)

Lines 132-133

Point 20: “We assessed disease severity (measured by the percentage of leaf area affected) as per Fehr et al. [24].” >  it is fine to insert bibliographic references of the methods but it is not pleasant not to have even a hint of the method itself.  It would also be appropriate to describe the assessment of both disease severity and disease incidence. different parameters but equally useful to describe the effect of a fungicide.

Response 20: The description of the method is in line 132-141. The sentence “We used a modified ‘X’ sampling pattern to select the plants and evaluated the disease incidence. Disease incidence was calculated using the formula D1/D2 × 100, D1 and D2 represent the number of diseased plants and surveyed plants , respectively” is added. (Line159-162 )

Line 150

Point 21: “106” > pay attention how the number is written.

Response 21: “106”has been changed to 106   (Line170 )

Line 154

Point 22: “disease spots” > it is better to describe the symptom

Response 22:“disease spots” has been changed to the “symptom” (Line175 )

Lines 155-156

Point 23: “(National Institute of Health, Bethesda, MD, USA).”  > this should be described in the methods not in te result’s chapter.

Response 23: “(National Institute of Health, Bethesda, MD, USA).” It has been removed. (Line175 )

Line 157

Point 24: “amount” > number, size or both?

Response 24: The amount describes the size. The sentence has changed to “Isolate 022ZW from the PDA plate caused the largest size of spots”. (Line176 )

Line 158

Point 25: “app.” > ?

Response 25: “app.” has been removed. (Line178 )

Line 162

Point 26:  “obverse” > “reverse”

Response 26: “obverse” has been changed to “reverse”  (Line181 )

Lines 181-183

Point 27: “We amplified the ITS, GAPDH, and TUB2 sequences using PCR ……. software.” > metods not results.

Response 27: “We amplified the ITS, GAPDH, and TUB2 sequences using PCR. Then, a multigene phylogenetic tree was constructed using the maximum likelihood (ML) method in MEGA 7.0 software” has been removed.  (Line194 )

Line 189

Point 28:” The accession numbers of isolate…” > The accession numbers are retated to sequences of isolates…

Response 28: ITS (OP523978), tef-1α (OP539309), and β-tub (OP539308) has been changed to ITS (OP523978),GAPDH (OP539309), and TUB(OP539308). (Line208 )

Line 20

Point 29: “C. fructicola” > when the text is in italics, latin name should be written normally.

Response 29: “C. fructicola” has been changed to Colletotrichum fructicola. (Line218 )

Lines 207-208

Point 30: why repeat these sentences?

Response 30: “The concentrations of honokiol, magnolol, carvacrol, thymol, citral, citronellol, and geraniol were 21.70, 24.04, 31.04, 31.97, 41.17, 58.11, and 59.96 µg/mL, respectively.” has been removed. (Line218 )

Lines 209-211

Point 31:  this sentences lack of something….. these are the concentration that showed a protection or the used? It should be specify.

Response 31: These concentration showed a treatment. Honokiol and magnolol shown good treatment effects, whereas those of carvacrol, thymol, citral, citronellol, and geraniol were less effective. (Line231 )

Line 241

Point 32: “Weconducted” > “We conducted”

Response 32: “Weconducted”  has been changed to “We conducted”  (Line262 )

Lines 249-250

Point 33: “There have been no reports on diseases caused by Colletotrichum fructicola in sorghum.” > It is useless: the same concept was illustraded the phrase before.

Response 33: “There have been no reports on diseases caused by Colletotrichum fructicola in sorghum.” has been removed.  (Line269 )

Lines 253-261

Point 34: “For example……” > it is better to cite the use of fungicides on sorghum.

Response 34: These have been modified. carbendazim and propiconazole could prevent anthracnose, grey leaf spot and zonate leaf spot of sorghum, and carbendazim also could controled wet root rot on sorghum [41], carbendazim was effectively inhibited the hypha of Colletotrichum fructicola [42], and Colletotrichum fructicola were highly sensitive to pyraclostrobin, difenoconazole, fludioxonil, tebuconazole, pyrisoxazole, and tetramycin in terms of mycelial growth inhibition[43]. Usman et al. study and find that long-term use of procymidone and fludioxonil can lead to increased resistance [45]. Chechi et al. found that if these fungicides are not applied scientifically, they can induce resistance in Colletotrichum fructicola. (Line273-285 )

Point 35: Table 4: Control percentage(%)> it is obvious that a control percentage was expressed in percentage > remove parenthesis

Response 35: “()”> has been removed.(Line250 )

Point 36:References: the titles of cited articles should be writte according the same rules….. uppercase/lowercase for the initials letters….

Response 36: I had  modified

Point 37: Be careful to the many latin names e.g. line 303 “pestalotiopsis trachycarpicola  >  Pestalotiopsis trachycarpicola  (also in italics),

Response 37: “pestalotiopsis trachycarpicola” has been changed to “pestalotiopsis trachycarpicola” (Line329 )

Point 38: also the journals…. Some in italics some not…. Please follow the instruction for the Authors. Also, in some cases, the English text should be revised.

Response 38: All journals has been italicized, and the English text has been revised.

Round 2

Reviewer 2 Report

Some suggestions to improve the manuscript:

After having written at the beginning of the text that the search is focused on Colletotrichum fructicola, in the rest of the manuscript the name of the mushroom can be abbreviated to C. fructicola.

Line 36: “ ha.” > “ha” units of measure have no period

Line 64 and 75: “Phytochemigals” > “phitochemicals” I apologize for my typo!

Lines 70-71: “(PDA: 200 g potatoes/L, 20 g glucose/L, 20 g agar/L, and 1 L distilled water)” > if 1 liter of water is specified in the recipe, all the previous doses must not refer to "/L" but it is sufficient to indicate the number of grams.. However in the PDA medium the infusion of 200 grams of potatoes is used and not 200 grams of potatoes as they are.

Line 73: “with” > “in”

Line 119: “ (mm)     > it is not necessary to indicate this unit of measurement in the materials and methods

Line 123: if the title of the paragraph is foreseen in italics, the Latin names must be written normally, not in italics

References: The titles of the scientific papers cited: either have all the initials of each word in capital letters or all with the initials in lower case. You have to standardize in relation to the rules of the journal.

Furthermore, it is necessary to verify all the Latin names contained: many are not written in italics. Authors also check for spaces between words: many are missing.

Line 137: “sympotms  cuaseed” > “symptoms caused” ?

Line 156: “Colletotrichum fructicola” > italics

Line 322: “pestalotiopsis “ > “Pestalotiopsis

Line 370: “Argentina,Crop“ > “Argentina, Crop

Line 371: “Glycine max” > italics.

Line 393: “China,Crop Prot,” > “China, Crop Prot,”    and delete “Volume”

Line 401: “Crop Prot,2013,47(Null)” > “Crop Prot, 2013, 47(Null)”

Author Response

Dear reviewer

Thank you for your valuable comments on my article. The Manuscript ID is jof-2181680. The title of this manuscript is First Report on Colletotrichum fructicola Causing Anthracnose in Chinese Sorghum and its management using phytochemicals.” I have answered all your comments. In addition, I carefully read the manuscript several times and found and corrected several minor errors. Added some content in the background section

In the rest of the manuscript the name of the mushroom have been abbreviated to C. fructicola.

Line 36:

Point 1: “ ha.” > “ha” units of measure have no period

Response 1: . has been deleted(Line 36)

Line 64 and 75:

Point 2: “Phytochemigals” > “phitochemicals” I apologize for my typo!

Response 2: It doesn't matter.

Lines 70-71:

Point 3: “(PDA: 200 g potatoes/L, 20 g glucose/L, 20 g agar/L, and 1 L distilled water)” > if 1 liter of water is specified in the recipe, all the previous doses must not refer to "/L" but it is sufficient to indicate the number of grams.. However in the PDA medium the infusion of 200 grams of potatoes is used and not 200 grams of potatoes as they are.

Response 3: “(PDA: 200 g potatoes/L, 20 g glucose/L, 20 g agar/L, and 1 L distilled water)” has been changed to “(PDA: potato infusion 200 g,  glucose 20 g , agar 20 g, and distilled water 1 L )” (Line 70-72)

Line 73

Point 4: “with” > “in”

Response 4: “with” has been changed to “in” (Line 73)

Line 119

Point 5: “ (mm)  “   > it is not necessary to indicate this unit of measurement in the materials and methods

Response 5: “(mm)”  has been deleted (Line 119)

Line 123

Point 6: if the title of the paragraph is foreseen in italics, the Latin names must be written normally, not in italics

Response 6: “Colletotrichum fructicola”  has been chenged to “Colletotrichum fructicola” (Line 123)

Point 7:References: The titles of the scientific papers cited: either have all the initials of each word in capital letters or all with the initials in lower case. You have to standardize in relation to the rules of the journal.

Response 7: I have standardized in relation to the rules of the journal.

Point 8: Furthermore, it is necessary to verify all the Latin names contained: many are not written in italics. Authors also check for spaces between words: many are missing.

Response 8: I have carried out a thorough inspection and revision (Line 17, 119, 135, 154, 137,142, 273, 302, 370, 374, 398, 406,)

Line 137

Point 9: “sympotms  cuaseed” >“sympotms  cuased”?

Response 9: “sympotms  cuaseed” has been changed to “sympotms  caused” (Line 137)

Line 156

Point 10: “Colletotrichum fructicola” > italics

Response 10: “Colletotrichum fructicola” has been changed to “Colletotrichum fructicola” (Line 156)

Line 322

Point 11: “pestalotiopsis “ > “Pestalotiopsis 

Response 11: “pestalotiopsis” has been changed to “Pestalotiopsis ” (Line 322)

Line Line 370

Point 12: “Argentina,Crop“ > “Argentina, Crop

Response 12: “Argentina,Crop” has been changed to “Argentina, Crop” (Line 370)

Line 371

Point 13: “Glycine max” > italics.

Response 13: “Glycine max” has been changed to “Glycine max” (Line 371)

Line 393

Point 14: “China,Crop Prot,” > “China, Crop Prot,”    and delete “Volume”

Response 14: “China,Crop Prot,” has been “China, Crop Prot,”  ï¼ˆLine 398)

Line 401

Point 15: “Crop Prot,2013,47(Null)” > “Crop Prot, 2013, 47(Null)”

Response 15: “Crop Prot,2013,47(Null)” has been changed “Crop Prot, 2013, 47(Null)” (Line 406)

I added two relevant references (Line 59-64)

Other changes are highlighted in red in the article(Line 24, 32,46,73,77,86,116, 171, 216, 226, 247, 252, 269,)